# Mechanical and Morphological Properties of Bio-Phenolic/Epoxy Polymer Blends

**DOI:** 10.3390/molecules26040773

**Published:** 2021-02-03

**Authors:** Ahmad Safwan Ismail, Mohammad Jawaid, Norul Hisham Hamid, Ridwan Yahaya, Azman Hassan

**Affiliations:** 1Laboratory of Biocomposite Technology, Institute of Tropical Forestry and Forest Products (INTROP), Universiti Putra Malaysia, UPM, Serdang 43400, Selangor, Malaysia; ahmadsafwanismail@gmail.com (A.S.I.); h_noroul@upm.edu.my (N.H.H.); 2Department of Forest Production, Faculty of Forestry, Universiti Putra Malaysia, UPM, Serdang 43400, Selangor, Malaysia; 3Science and Technology Research Institute for Defence, Kajang 43000, Selangor, Malaysia; adr266@gmail.com; 4Faculty of Engineering, School of Chemical and Energy Engineering, Universiti Teknologi Malaysia, Skudai, Johor Bharu 81310, Malaysia; azmanh@cheme.utm.my

**Keywords:** epoxy, bio-phenolic, polymer blends, tensile properties, impact properties, flexural properties, morphological properties

## Abstract

Polymer blends is a well-established and suitable method to produced new polymeric materials as compared to synthesis of a new polymer. The combination of two different types of polymers will produce a new and unique material, which has the attribute of both polymers. The aim of this work is to analyze mechanical and morphological properties of bio-phenolic/epoxy polymer blends to find the best formulation for future study. Bio-phenolic/epoxy polymer blends were fabricated using the hand lay-up method at different loading of bio-phenolic (5 wt%, 10 wt%, 15 wt%, 20 wt%, and 25 wt%) in the epoxy matrix whereas neat bio-phenolic and epoxy samples were also fabricated for comparison. Results indicated that mechanical properties were improved for bio-phenolic/epoxy polymer blends compared to neat epoxy and phenolic. In addition, there is no sign of phase separation in polymer blends. The highest tensile, flexural, and impact strength was shown by P-20(biophenolic-20 wt% and Epoxy-80 wt%) whereas P-25 (biophenolic-25 wt% and Epoxy-75 wt%) has the highest tensile and flexural modulus. Based on the finding, it is concluded that P-20 shows better overall mechanical properties among the polymer blends. Based on this finding, the bio-phenolic/epoxy blend with 20 wt% will be used for further study on flax-reinforced bio-phenolic/epoxy polymer blends.

## 1. Introduction

Currently, the needs of high-performance materials have driven researchers to studies on various methods, such as polymer synthesis and modification. There are several techniques in polymer modification such as blending, copolymerization, and reinforcing [1]. Polymer synthesis and polymer blends can produce a new material. However, polymer synthesis is an expensive method as compared to polymer blends, which is a cheap and easy method. Moreover, polymer blends have better process ability, quick formulation changes, tailor ability to a specific needed, reduced number of grades need to manufacture and store, and recyclability of blends achieved by control of morphology [2]. Through polymer blending, two or more polymers are combined in a single system, which produce a new material with unique properties [3]. The blending can be either from the same or different categories.

Several research studies have been reported in polymer blending using various combinations of polymers. Chen et al. [1] has studies of blending thermoplastic with thermoplastic polymers. In this study, a different loading of polyamide (PA) is a blend with polyphenylene sulfide (PPS). The mechanical properties of blends were improved and found that there is a two-phase structure in the blends. A study in thermoplastic with a thermoset polymer was reported by Mishra et al. [4]. Lignin and polyvinyl chloride were used, and it was found that the film produced does not show any sign of phase separation. Unnikrishnan and Tachil [5] has studies on blending epoxy resin with epoxidized novolacs. The epoxidized material was prepared with a different stoichiometry of phenol and formaldehyde (1:0.6, 1:0.7. 1:0.8, and 1:0.9). It was found that blending epoxy with epoxidized novolac improve the mechanical properties. In addition, the Unnikrishnan and Tachil [6] study on blending epoxy with phenol and substituted phenol such as p-cresol, t-butyl phenol, and cardanol. It was found that blending epoxy with phenol and substituted phenol has improved the mechanical properties. 

Mohamed et al. [7] has a study on toughening epoxy with polysulfide rubber (PSR). It is shown that addition of PSR increase the impact resistance and elongation at break while tensile strength and the modulus decreased. Another research study on epoxy polymer blends was studied by Sun et al. [8]. In this study, polysulfone (PSF) was used to toughen and modify epoxy. There is a significant improvement in fracture toughness and impact strength of PSF/epoxy blends compared to neat epoxy. However, there are phased separations between PSF and epoxy resin, which are shown in scanning electron microscopy. Epoxy blending with polyetherketone cardo was studied by Zhou et al. [9]. Blend epoxy with polyetherketone cardo has improved tensile properties, flexural properties, impact strength, and fracture toughness. Phase separation between polyetherketone cardo and epoxy was detected in scanning electron microscopy.

The selection of the polymer to blend depends on its application and the drawback of the polymer that needs improvement. Phenolic resin was used in many areas, such as molding compound, wood products industry, foundry, composite materials, structural integrity, thermal insulation materials, coating, thermal stability, and solvent resistance [10,11,12,13,14,15]. Phenolic resin can be used in various applications due to its properties such as excellent dimensional stability at high temperatures, excellent thermal properties, excellent flame resistance, heat insulation properties, highly crosslinked thermoset, high rigidity efficiency in glue-bond formation, exceptional adhesive properties, and chemical stability [16,17,18]. There are two types of phenolic resin, which is petroleum and bio-based phenolic resin. Currently, petroleum was the main source for the production of phenolic resin. However, there are efforts to replace petroleum with biomass as sources of raw materials for producing phenolic resin. There are three reasons for replacing petroleum-based phenolic with bio-based phenolic, which are needed for more sustainable bio-sources of materials, enhance the performance of the resin, and improve health and safety during manufacturing and end use [19]. Epoxy is another thermoset polymer, which have excellent mechanical properties compared to phenolic. Epoxy is a well-known polymer and has been used in many high-performance applications such as structural application, aerospace industry, and body armor [20,21]. In comparison, epoxy has excellent mechanical properties while phenolic has excellent thermal stability and resistance. In order to produce a material that has balance properties in thermal and mechanical properties, blending of epoxy and phenolic in a single system can be considered. 

In this study, bio-phenolic resin was blended with epoxy resin and both were commercially available. Therefore, there is no issue regarding the supply for both resins for industrial usage. The purposed of this study is to investigate the effect of different loadings of bio-phenolic resin on tensile, flexural, impact, and morphological properties of epoxy composite. Based on literature, it was expected that the mechanical properties of bio-phenolic/epoxy polymer blends were improved compared to neat epoxy and phenolic composites. The finding in this study will be used in a future study for ballistic helmet application by using flax and carbon/kevlar as reinforcement.

## 2. Materials and Methods

### 2.1. Materials

Epoxy resin D.E.R * 331 and epoxy hardener jointmine 905-3S was procured from Tazdiq Engineering Sdn Bhd. (Selangor, Malaysia). While Bio-phenolic (PH-3507) was obtained by Chemovate Girinagar, Banglore, India. The Teflon sheet was supplied by Evergreen Sdn Bhd. (Kuala Lumpur, Malaysia). Table 1, Table 2 and Table 3 showed the properties of epoxy resin D.E.R * 331, Jointmine 905-3S, and Bo-phenolic (PH-3507). Figure 1 showed the chemical structure for the materials.

### 2.2. Fabrication of Polymer Blends

Epoxy resin was stirred using mechanical stirring and, at the same time, bio-phenolic resin was added slowly to avoid bio-phenolic resin agglomerate. The mixture was stirred for about 10 min so that the powder was evenly distributed in the epoxy. Then epoxy hardener was poured into the mixture and the stirring continued for about another 2–4 min. This mixing process was done at room temperature. A Teflon sheet was placed in between the mould and steel plate. The mixture was poured into the stainless steel mould with dimensions of 150 mm × 150 mm × 3 mm. The mould was transferred into the hot press with a temperature of 150 °C for 15 min. After 15 min, the polymer blend was removed from the mould and cooled at room temperature. The polymer blends were put in the oven at 105 °C for 2 h. The formulation of the polymer blends was shown in Table 4.

### 2.3. Characterization

#### 2.3.1. Tensile Properties

Tensile testing was conducted as per ASTM D 3039 using a 30 kN Bluehill INSTRON 5567 universal testing machine (Shakopee, MN, USA). The samples were cut using bandsaw JETMAC JMWBS-14 with dimension 120 mm × 20 mm × 3 mm. The testing speed used was 2 mm/min and 60-mm gauge length was used. The sample was put in a condition chamber for 24 h at 23 ± 3 °C and a relative humidity of 50 ± 10%. In every formulation, five replications were tested, and the average value was tabulated.

#### 2.3.2. Flexural Properties

Flexural testing was conducted as per ASTM D 790 using a 30 kN Bluehill INSTRON 5567 universal testing machine (Shakopee, MN, USA). The samples were cut using a bandsaw JETMAC JMWBS-14 with dimension 127 mm ×12.7 mm × 3 mm. The support span used was 16 times the depth of the sample and the crosshead speed used was calculated using Equation (1). The sample was put in a condition chamber for 24 h at 23 ± 3 °C and a relative humidity of 50 ± 10%. In every formulation, five replications were tested, and the average value was tabulated.
R = 0.01 L^2^/6d,(1)
where:

R = rate of crosshead motion, mm/min,

L = support span, mm,

d = depth of beam, mm

#### 2.3.3. Impact Properties

Impact testing was conducted as per ASTM D 256 using a Ray Ran advanced universal pendulum impact tester (RR/IMT) (Nuneaton, UK) The samples were cut using bandsaw JETMAC JMWBS-14 with dimension 63.5 mm ×12.7 mm × 3 mm. In every formulation, five replications were tested, and the average value was tabulated.

#### 2.3.4. Scanning Electron Microscopy (SEM)

The fracture surface of polymer blends, epoxy, and phenolic from the tensile test were examined using an EM-30AX scanning electron microscope (COXEM, Daejeon, Korea) with an acceleration voltage of 20 kV. The samples were coated with a thin layer of gold prior to the testing.

## 3. Results and Discussion

### 3.1. Tensile Properties

The ability of bio-phenolic/epoxy polymer blends to resist breaking under tensile stress and its rigidity were studied by examining the tensile strength and modulus. Figure 2 shows the effect of bio-phenolic loading on tensile strength of epoxy. The obtained results showed that neat epoxy have higher tensile strength compared to the bio-phenolic, which is 53 MPa and 32 MPa, respectively. It also clears from results that the addition of 5 wt% bio-phenolic slightly reduce the properties of the polymer blend compared to neat epoxy while it has better tensile strength compared to neat phenolic. Decreases in tensile strength of P-5 compared to neat epoxy might be due to the interaction between bio-phenolic and epoxy resin at 5 wt% of bio-phenolic, which is not satisfactory. However, a further increase in bio-phenolic resin loading from 10 wt% to 25 wt% showed improvement in tensile strength compared to both neat epoxy and bio-phenolic material. Increases in mechanical properties of phenolic/epoxy polymer blends was due to the higher degree of cross-linking, chain extension, and some amount of entangling of the polymer chain [6]. This indicates that the interaction between epoxy and bio-phenolic material has been improved. As a result, it will reduce the stress concentration point when tensile load is applied to the polymer blends as a result of the tensile strength increase [24,25]. Addition of 25 wt% of bio-phenolic slightly decrease tensile strength compared to 20 wt% bio-phenolic material. The highest improvement was shown by P-20 (66 MPa), which is 1.27 times compared to epoxy material (52 MPa). 

The effect of bio-phenolic material on the tensile modulus of epoxy were shown in Figure 3. The control polymer showed that bio-phenolic has 1.8 times higher tensile modulus compared to epoxy material. Moreover, blends of bio-phenolic/epoxy material have improved the tensile modulus compared to neat epoxy but have not exceeded the tensile modulus of neat bio-phenolic material. The data showed that an increase in the weight percentage of bio-phenolic in epoxy from 5 wt% to 25 wt% has increased the tensile modulus. Bio-phenolic material was able to make epoxy stiffer as the tensile modulus increased with the addition of bio-phenolic material. This attribute due to the properties of bio-phenolic material have a high tensile modulus. The lowest and highest tensile modulus among the polymer blends are P-5 and P-25, which is 3.1 GPa and 3.7 GPa, respectively. The improvement in tensile modulus shown by P-5 and P-25 is about 6.48% and 25.94% compared to neat epoxy material.

Scanning electron microscopy was carried out on a fracture surface of the tensile test. Figure 4 showed a tensile fracture surface of epoxy, bio-phenolic material, and bio-phenolic/epoxy polymer blends. Based on Figure 4a,b, the fracture surface of epoxy (Figure 4a) showed a multilevel fracture with ridges and a wavy crest while fracture surface paths of bio-phenolic material (Figure 4b) mostly showed straight and constitute failure bands. This indicates that there are more energy absorptions at a large scale during failure for epoxy compared to bio-phenolic material. According to Unnikrishnan and Tachil, a multilevel fracture with ridges and a wavy crest indicate there are high energy absorption during failure [6]. Thus, it agreed with the finding which showed neat epoxy has higher tensile strength compared to bio-phenolic material. Figure 4c–g showed the fracture surface of bio-phenolic/epoxy polymer blends with different loadings of bio-phenolic material. It is shown that the degree of roughness in bio-phenolic/epoxy polymer blends seem to increase as bio-phenolic loading increase. In addition, a multilevel fracture with ridges and wavy crest is observed. This finding is aligned with the result, which showed an increase in tensile strength as bio-phenolic loading increase. Moreover, the surface fracture of bio-phenolic/epoxy polymer blends does not show any sign of phase separation. Micrograph fracture of all samples are comparable, which is only different on the degree of roughness. This indicates that the blends are homogeneous and there are fully co-crosslinked polymers networks in the blends [26,27].

### 3.2. Flexural Properties

The ability of bio-phenolic/epoxy polymer blends to withstand bending forces perpendicular to its longitudinal axis, which resist to bending forces, were studied by evaluating the flexural strength and modulus. Figure 5 showed the effect of different bio-phenolic loading on the flexural strength of epoxy. The flexural strength of polymer blends was increased as bio-phenolic loading increased from 5 wt% to 20 wt% whereas it slightly decreased in flexural strength, which was seen at 25 wt% bio-phenolic (116 MPa) compared to 20 wt% bio-phenolic material (120 MPa). The findings showed that bio-phenolic/epoxy polymer blends have higher flexural strength compared to neat epoxy and bio-phenolic material. This attribute was due to the formation of a network structure between two polymers that improved the flexural strength of polymer blends [28,29]. Besides that, improvement in adhesion between bio-phenolic and epoxy material has increased stress transfer in the polymer blends as a result of higher flexural strength.

Figure 6 showed the effect of bio-phenolic loading on the flexural modulus of epoxy material. Based on finding the flexural modulus of bio-phenolic/epoxy polymer blends are lower compared to neat bio-phenolic material. This is mainly due to the lower flexural modulus of epoxy which is the main polymer in the blends. However, flexural modulus of bio-phenolic/epoxy blends was improved compared to neat epoxy material. The trend of flexural modulus is increased as bio-phenolic loading increased up to 10 wt% and slightly decreased at 15 wt% bio-phenolic material. However, the trend continues to increase as bio-phenolic loading increased from 15 wt% to 25 wt%. Bio-phenolic/epoxy polymer blends at 25 wt% showed the highest flexural modulus (3.9 GPa), which is a 27.45% improvement compared to neat epoxy. The increase in flexural modulus due to cross-linking between bio-phenolic and epoxy material as a result of rigidity of polymer blends increase. 

### 3.3. Impact Properties

The ability of a material to absorb and dissipate energy under impact or shock loading were studied by evaluating the impact property [30]. Figure 7 showed the impact of blending of bio-phenolic material with epoxy material. There are no significant differences in impact resistance for neat epoxy and bio-phenolic material. The addition of 5 wt% bio-phenolic in epoxy polymer gave a negative effect on the impact resistance and it is lower compared to both neat epoxy and bio-phenolic material. The impact resistance of P-5 was decreased by about 41.94% compared to epoxy material. A decrease in impact resistance might be due to the weak interaction between bio-phenolic and epoxy material. A similar trend was shown in tensile strength for P-5. However, a further increase in bio-phenolic loading seems to have a positive effect on impact resistance of bio-phenolic/epoxy polymer blends. The impact resistance gradually increases as bio-phenolic loading increases from 10 wt% to 20 wt%. In addition, a further increase of bio-phenolic loading from 20 wt% to 25 wt% does not show a significant difference in impact resistance. The improvement in impact resistance was due to improvement in the interaction between bio-phenolic and epoxy material. Besides that, formation of the network structure between bio-phenolic and epoxy material, higher degree of cross-linking, chain extension, and some amount of entangling of the polymer chain is also the reason for an impact resistance increase. This helps the materials to generate plastic deformation zones along the fracture direction to absorb more energy as a result of impact resistance of the blends, which improved [31,32]. According to Wang et al. [33], improvement in interfacial adhesion and entanglement of the molecular chain helps the blends to absorb more energy when damaged. The highest impact resistance was shown by a bio-phenolic/epoxy polymer blend with 20 wt%, which is 18 J/m. 

## 4. Conclusions

The highest tensile strength improvement was shown by P-20 (66 MPa), which is 1.27 times compared to epoxy (52 MPa). The obtained results showed that neat epoxy have higher tensile strength compared to bio-phenolic material, which is 53 MPa and 32 MPa, respectively, whereas P-5 display the lowest (3.1 GPa) and P-25 show the highest tensile modulus (3.7 GPa). A morphological study of tensile fracture samples supported this finding based on SEM images showing a multilevel facture with ridges and a wavy crest while fracture surface paths of a bio-phenolic material mostly showed straight and constitute failure bands. The findings showed that bio-phenolic/epoxy polymer blends have higher flexural strength compared to neat epoxy and bio-phenolic material. Bio-phenolic/epoxy polymer blends at 25 wt% showed the highest flexural modulus (3.9 GPa), which is 27.45% improvement compared to neat epoxy material. The highest impact resistance (18 J/m) was shown by a bio-phenolic/epoxy polymer blend with 20 wt%. The findings showed that a bio-phenolic/epoxy polymer blend has potential to be utilized for an application since it showed a positive result. Based on this result, P-20 showed the best overall properties. We concluded from this study that blends of bio phenolic resin with epoxy resin can improve the overall mechanical properties of bio phenolic/epoxy polymer blends. Thus, bio-phenolic/epoxy polymer blends with 20 wt% bio-phenolic material will be used in a future study for a ballistic helmet application by using flax and carbon/Kevlar as reinforcement.

## Figures and Tables

**Figure 1 molecules-26-00773-f001:**
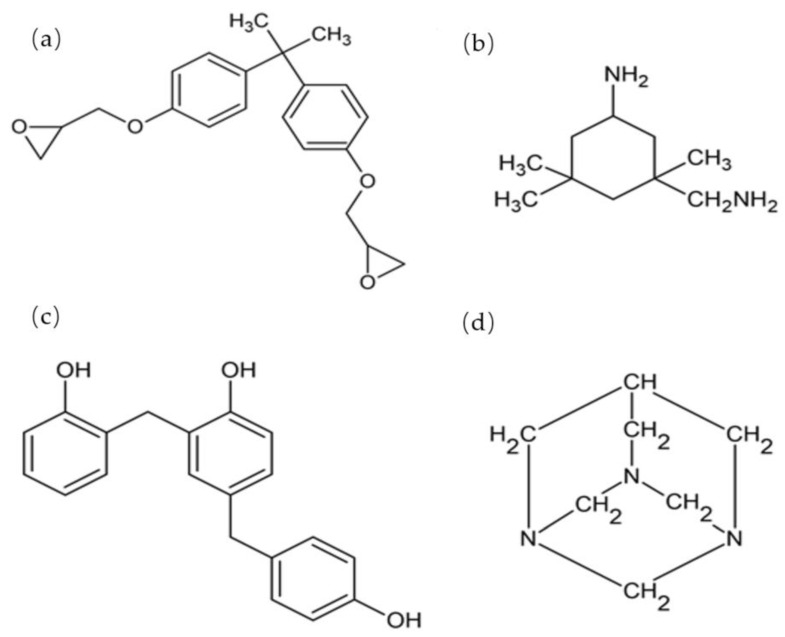
(**a**) Epoxy D.E.R * 331 [22], (**b**) Jointmine 905-3S, (**c**) Bio-phenolic (PH-3507) [23], (**d**) Hexamine [18].

**Figure 2 molecules-26-00773-f002:**
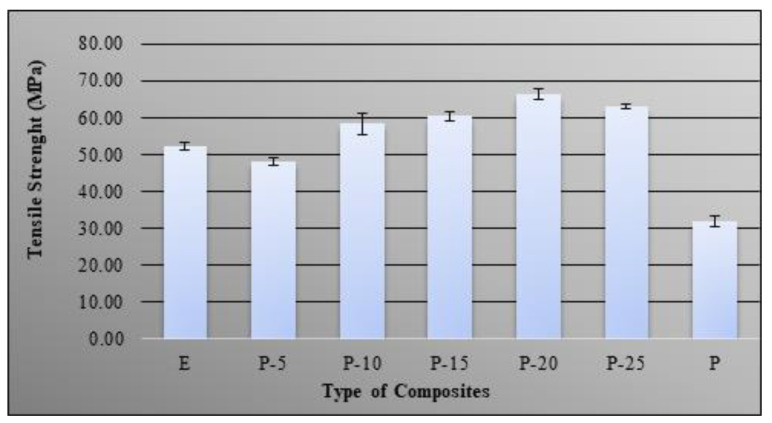
Tensile strength of bio-phenolic/epoxy blends, neat epoxy, and bio-phenolic.

**Figure 3 molecules-26-00773-f003:**
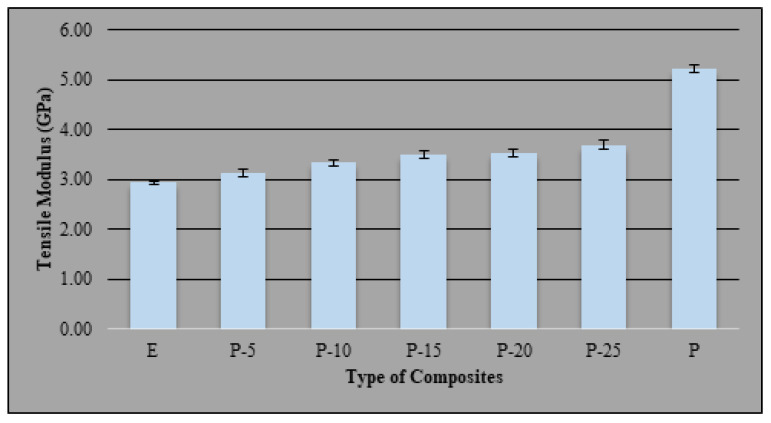
Tensile modulus of bio-phenolic/epoxy polymer blends, neat epoxy, and bio-phenolic material.

**Figure 4 molecules-26-00773-f004:**
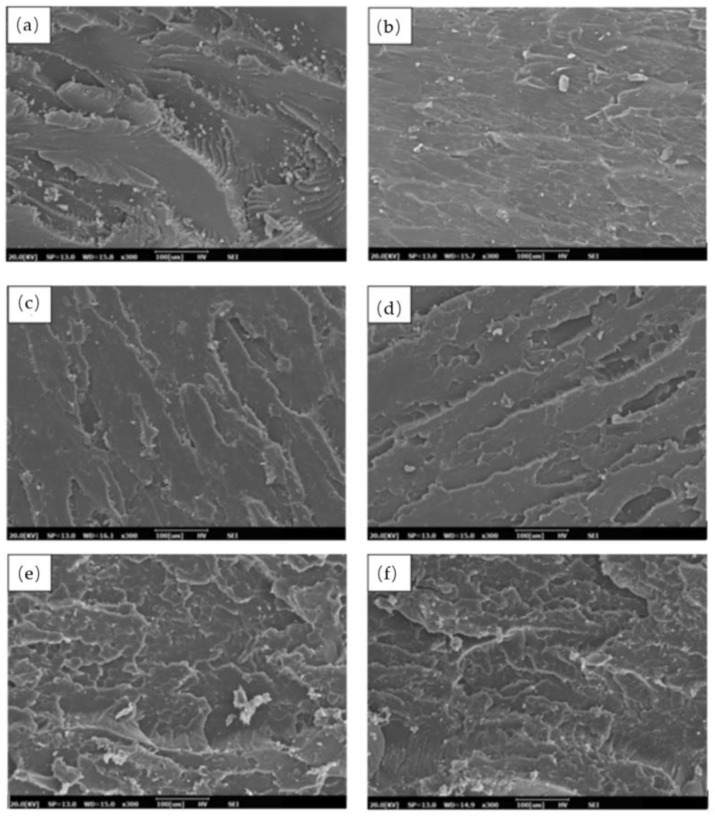
SEM of tensile fractured: (**a**) epoxy (**b**) bio-phenolic (**c**) P-5 (**d**) P-10 (**e**) P-15 (**f**) P-20 (**g**) P-25.

**Figure 5 molecules-26-00773-f005:**
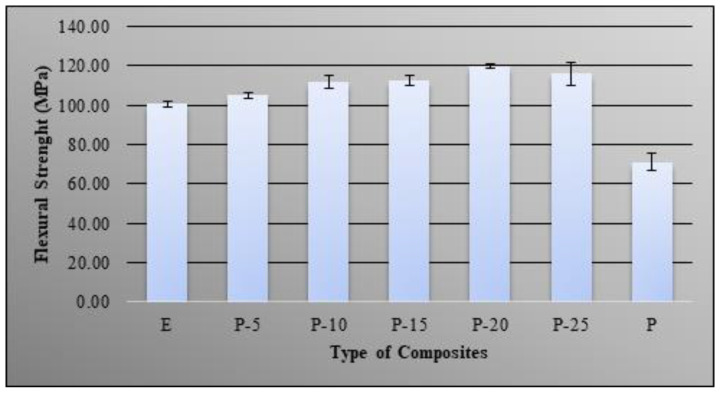
Flexural strength of bio-phenolic/epoxy blends, neat epoxy, and bio-phenolic material.

**Figure 6 molecules-26-00773-f006:**
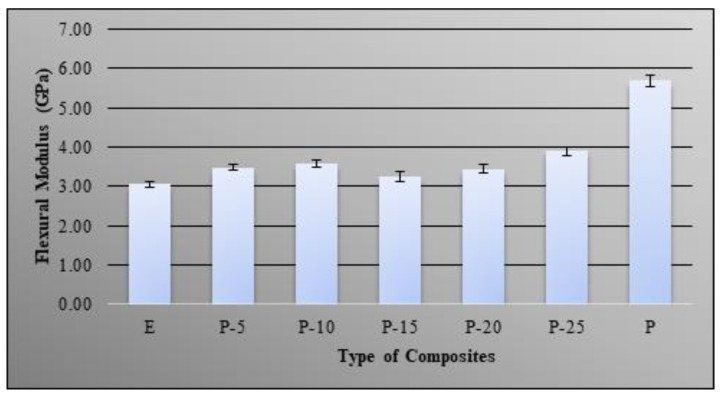
Flexural modulus of bio-phenolic/epoxy blends, neat epoxy, and bio-phenolic material.

**Figure 7 molecules-26-00773-f007:**
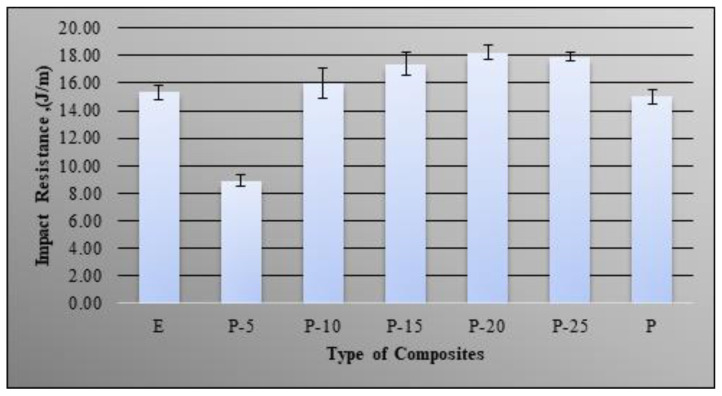
Impact resistance of bio-phenolic/epoxy blends, neat epoxy and bio-phenolic.

**Table 1 molecules-26-00773-t001:** Properties of epoxy resin D.E.R * 331.

Property	Value
Epoxide Equivalent Weight (g/eq)	182–192
Epoxide Percentage (%)	22.4–23.6
Epoxide Group Content (mmol/kg)	5200–5500
Color (Platinum Cobalt)	75 Max.
Viscosity at 25 °C (mPa·s)	11,000–14,000
Hydrolyzable Chloride Content (ppm)	500 Max.
Water Content (ppm)	700 Max.
Density at 25 °C (g/mL)	1.16
Epichlorohydrin Content (ppm)	5 Max.
Shelf Life (Months)	24

**Table 2 molecules-26-00773-t002:** Properties of Jointmine 905-3S.

Property	Value
Amine value (mg KOH/g)	300 ± 20
Viscosity (BH type at 25 °C, cPs)	200∼400
Color (Gardner)	<2
Equivalent Wt (H)	95
Pot life (100 g at 25 °C)	75 min
Hardness (Shore D)	85
Thin film set time (at 25 °C)	5 h

**Table 3 molecules-26-00773-t003:** Properties of Bio-phenolic (PH-3507).

Property	Description
Physical state	Powder
Colour	Reddish Brown
Odour	Slide odour of phenol
pH	8–8.5
Melting point by capillary	81 °C
Curing time at 150 °C with 10% hexa by a plate spatula method	82 s
Hexamine %	10.08%
Sieve analysis +200 mesh BS+240 mesh BS	0.72%
Free phenol	<1% strictly

**Table 4 molecules-26-00773-t004:** Formulation and label of each sample.

Label	Phenolic Resin (wt%)	Epoxy Resin (wt%)
E	0	100
P-5	5	95
P-10	10	90
P-15	15	85
P-20	20	80
P-25	25	75
P	100	0

## Data Availability

Not Applicable.

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
