# Peer review of "Mechanical and Morphological Properties of Bio-Phenolic/Epoxy Polymer Blends"

_molecules, 2021, doi:10.3390/molecules26040773_

Round 1
Reviewer 1 Report
The submitted manuscript presents a study on the mechanical and morphological properties of bio-phenolic/epoxy polymer blends.
The study seems to be far from the main Journal scopes. The presented manuscript needs major improvement to be considered for publication.
The language and writing style is not satisfactory. Rather require major improvements.
The authors have to check the manuscript carefully.
The introduction misses some information. I encourage authors to rewrite this section.
You have to specify excellent mechanical properties, etc.
You have to provide chemical structures of the materials.
Conclusions seem to be confusing. This section contains only obvious statements.
“The bio-phenolic/epoxy polymer blends with 20 wt% bio-phenolic will be used in the future study for ballistic helmet application by using flax and carbon/kevlar as reinforcement.”
There is no clear information on this aim in the article.
Line 157: "This indicates that the interaction between epoxy and bio-phenolic was strengthened."
The Authors have to discuss these interactions in detail.
Author Response
We revised manuscript as per your comments. attached detailed reply.

Reviewer 2 Report
The authors prepared one more blend of two polymers and measured its mechanical properties. One could estimate the work in this way if not to pay attention to the nature of the used components. Meanwhile, this is rather interesting that one of the components was a bio-product and it was mixed at a different proportion with epoxy resins.
I also have some concrete comments/
- I doubt that the majority of readers know what bio-phenolic resin is. So, I am sure that this material should be described rather carefully.
- Almost the same comment touches the second component which is called “Epoxy resin D.E.R * 331 and epoxy hardener jointmine 905-3S.” If the paper is addressed to a wide auditorium not to local consumers only, the chemistry of both, resin and hardener should be deciphered.
- Figs 1 and 2 say nothing interesting and important for the sense of the work and should be deleted.
- There are no statistics concerning the results of the measurements. The authors write: “strength compared to bio-phenolic 150 composites is 53.17 MPa and 31.92 MPa.” It is much more correct to say: 53 and 32 MPa. The same can be said about modulus: not 3.12 GPa and 3.69 GPa but 3.1 and 3.7 GPa.
- The list of references (30 names) is too long for such a paper. Anyway, it is impossible to list all papers related to the theme of blends based on epoxy resin.
Author Response
We revised manuscript as per your comments and attached detailed reply

Round 2
Reviewer 1 Report
The revised manuscript meets the minimum standard. The Authors roughly responded to the comments. The writing style is far from good standards, and language problems need to be repaired. Despite this, the study seems to be plausibly valuable for further research in the field.
Reviewer 2 Report
Generally speaking, there is nothing bad in this paper.
However, I did not understand why the authors did not pay attention to my former comment:
The list of references (30 names) is too long for such a paper. Anyway, it is impossible to list all papers related to the theme of blends based on epoxy resin.
Instead of this, they increase the list of References up to 33.
I insist that the list of references should be halved.